# Molecular rheotaxis directs DNA migration and concentration against a pressure-driven flow

Sarah M. Friedrich [1], Jeffrey M. Burke[2], Kelvin J. Liu [2], Cornelius F. Ivory[3] & Tza-Huei Wang[1,4]

In-line preconcentration techniques are used to improve the sensitivity of microfluidic DNA analysis platforms. The most common methods are electrokinetic and require an externally applied electric field. Here we describe a microfluidic DNA preconcentration technique that does not require an external field. Instead, pressure-driven flow from a fluid-filled micro-capillary into a lower ionic strength DNA sample reservoir induces spontaneous DNA migration against the direction of flow. This migratory phenomenon that we call Molecular Rheotaxis initiates in seconds and results in a concentrated DNA bolus at the capillary orifice. We demonstrate the ease with which this concentration method can be integrated into a microfluidic total analysis system composed of in-line DNA preconcentration, size separation, and single-molecule detection. Paired experimental and numerical simulation results are used to delineate the parameters required to induce Molecular Rheotaxis, elucidate the underlying mechanism, and optimize conditions to achieve DNA concentration factors exceeding 10,000 fold.

[1] Biomedical Engineering Department, Johns Hopkins University, Baltimore, MD 21218, USA. [2] Circulomics, Inc., Baltimore, MD 21211, USA. [3] Gene and Linda Voiland School of Chemical Engineering and Bioengineering, Washington State University, Pullman, WA 99164, USA. [4] Mechanical Engineering Department, Johns Hopkins University, Baltimore, MD 21218, USA. Correspondence and requests for materials should be addressed to C.F.I. (email: cfivory@wsu.edu) or to T.-H.W. (email: thwang@jhu.edu)

Nucleic acid analysis is utilized in diverse fields ranging from healthcare to forensics to homeland security. The need for rapid and high sensitivity nucleic acid detection continues to grow for particular applications, where providing timely results with limited resources remains challenging. Microfluidic and single-molecule methods offer alternative solutions to achieve rapid and high sensitivity nucleic acid detection. Solution-based single-molecule detection strategies[1–3] enable high detection sensitivity without the need for amplification and are instead limited only by the volume of sample that can be efficiently screened. Such detection methods couple nicely to microfluidic analysis because microfeatures can be designed to match the fluidic analysis volumes with the single-molecule detection volume for highly efficient sample screening[4–6]. Beyond precise sample handling, scaling down to microfluidic scales offers additional benefits, including decreased sources of background signals (leading to increased sensitivity and specificity), increased reaction kinetics (decreasing assay time), and reduced sample and reagent usage (decreasing cost and requisite sample size)[7, 8]. In coupled microfluidic/single-molecule platforms, the assay sensitivity is limited by the number of molecules passed through the detector, which is ultimately determined by the sample concentration and the input sample volume.

One solution to further increase sensitivity in microfluidic systems is to preconcentrate the target prior to analysis. Evaporation and solid-phase extraction-based concentration methods have both been demonstrated in microfluidic devices[9–12], but electrokinetic methods are the most common and have demonstrated the highest nucleic acid concentration factors[13, 14]. Isotachophoresis (ITP) and field-amplified sample stacking (FASS) are routinely used in capillary electrophoresis (CE)[15] and can achieve nucleic acid concentration factors as high as 1000- to 10,000-fold[16, 17]. Despite the popularity of these electrokinetic methods, there remain several challenges to their implementation in microfluidic devices: namely the requirements of embedded electrodes, external power supplies, microchannel wall conditioning and treatment to reduce electroosmotic flow[18, 19], and design considerations that avoid electric-field-induced DNA aggregation and degradation[20].

It was during the development of an electrokinetic concentration method for capillary sampling that we observed that certain conditions spontaneously triggered dilute DNA to collect into a highly concentrated bolus even without an applied electric field. Under further study, we found that driving buffer flow out of the capillary actually caused DNA to migrate against the flow and gather even more quickly at the capillary inlet. We have

termed the phenomenon Molecular Rheotaxis (MRT), inspired by the biological rheotaxis, alignment and migration against a flow current, of fish[21], bacteria[22, 23] and sperm cells[24], and the "artificial" rheotaxis of engineered anisotropic particles[25, 26]. As this system does not require electrodes or wall coatings and uses simple buffer systems, it can be incorporated into nearly any microfluidic system or analytical chemistry platform (CE, high-pressure liquid chromatography, hydrodynamic chromatography, etc.), providing an elegant electrode-free method for DNA preconcentration.

Herein, we explore the underlying mechanism and concentration enhancement capabilities of this simple microfluidic preconcentration method through complementary experimental and numerical simulation approaches. First, a numerical model is used to pinpoint the critical experimental conditions and forces involved in the MRT concentration mechanism. Next, we investigate the effects of flow rate, time, and buffer conditions in order to further refine the mechanism and optimize the concentration enhancement. To simultaneously quantify the concentration enhancement over a wide range of DNA sizes, we couple MRT to a single molecule, free solution hydrodynamic separation platform (SML-FSHS). SML-FSHS uses only pressure-driven flow through micron sized capillaries to separate DNA fragments by size in free solution[27, 28]. Since SML-FSHS is performed using only a buffer-filled microchannel without sieving matrices or drag-tag conjugates to modulate mobility, wall coatings, or applied electric fields[27, 28], interfacing with MRT provides a truly electrode-free platform for highly sensitive and quantitative DNA preconcentration, size separation, and single-molecule detection. Both experimental and simulation results indicate that the technique is capable of effective DNA preconcentration with minimal size-dependent bias, demonstrating future applicability to a wide range of nucleic acid samples, assays, and applications. Finally, using optimized conditions with the fully integrated preconcentration and separation platform, we demonstrate concentration factors exceeding 10,000 fold with HindIII digested λ DNA from a starting concentration as low as 150 aM.

## Results

**Proposed concentration mechanism.** We unintentionally discovered MRT while observing the interface of a capillary orifice filled with a high ionic strength running buffer surrounded by a reservoir containing DNA in a low ionic strength sample buffer (Fig. 1). When pressure was used to drive flow of the high ionic

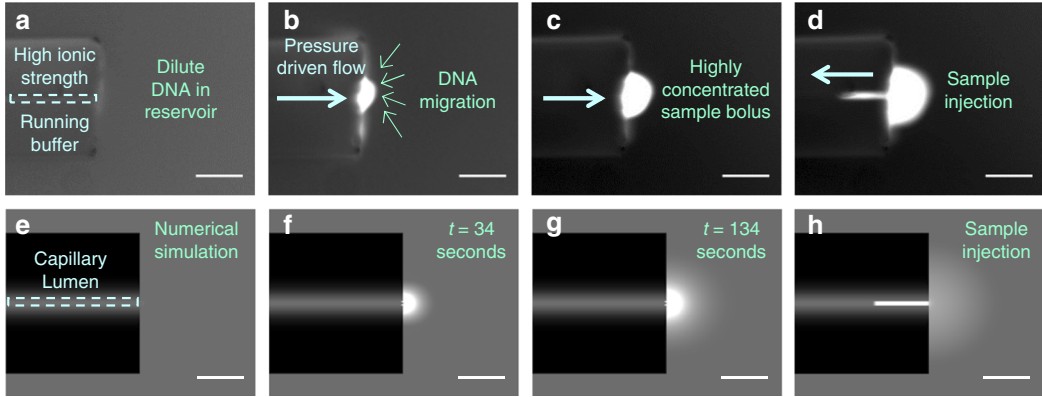

**Fig. 1** DNA preconcentration and injection into a microcapillary. **a–c** Snapshots capture DNA preconcentration at the outlet of a 5 μm inner diameter microcapillary when pressure is applied to drive flow of a high ionic strength solution out of the capillary and into the low concentration reservoir. **d** The bolus of concentrated DNA can be injected into the capillary by quickly reversing the flow direction. **e–h** The numerical simulation qualitatively captures the behavior of the DNA molecules in response to pressure-driven flow and the imposed buffer mismatch at the 5 μm capillary outlet. Scale bars are 50 μm

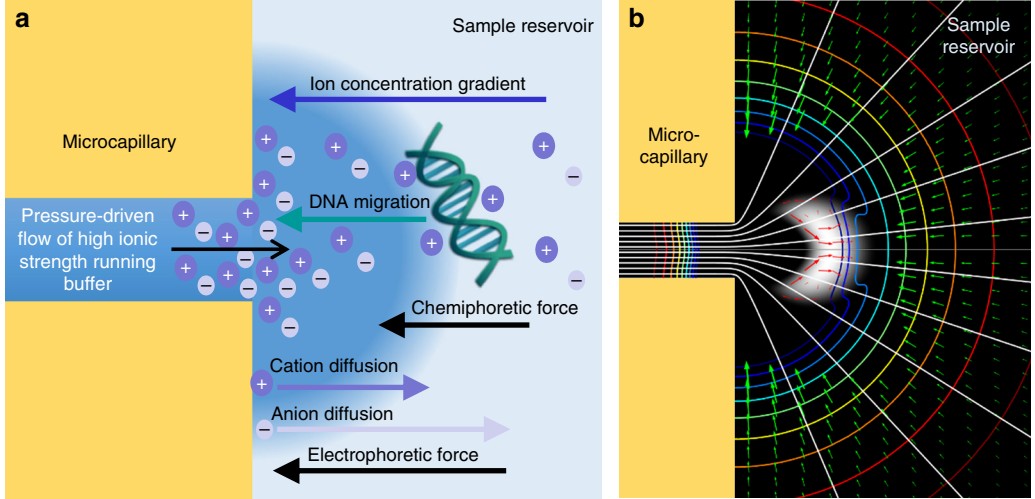

**Fig. 2** Mechanism of DNA Molecular Rheotaxis (MRT). **a** Schematic of the underlying mechanism of MRT. Pressure-driven flow drives the high ionic strength running buffer out of the capillary into the low ionic strength reservoir buffer. This forms a concentration gradient of ions surrounding the capillary orifice. The cations and anions migrate down their concentration gradients per their individual diffusivities. If anions diffuse faster than the cations, and induced electric field is generated that causes DNA to migrate toward the higher concentration of ions at the capillary orifice. Simultaneously, positive interactions between negatively charged DNA and cations can induce a chemiphoretic DNA migration towards the higher cation concentration near the capillary. **b** Results from the simulation show that flow out of the capillary (white streamlines emanating from microcapillary) is accompanied by an induced electric field (concentric rainbow contour lines) and results in DNA accumulation in a bolus near the capillary orifice (white shading). The direction and magnitude of DNA flux from the reservoir toward the bolus is shown in green arrows and with red arrows within the bolus. The original image was mirrored across the centerline to view the whole capillary (for the original figure, see Supplementary Fig. 1)

strength buffer out of the capillary, counter-intuitively, DNA within the reservoir migrated against the direction of flow and gathered into a highly concentrated bolus at the capillary orifice (Fig. 1c, Supplementary Movie 1). This concentration occurred despite the absence of an applied electric field. Then, by simply reversing the flow direction, we saw that the concentrated sample plug could be injected into the capillary (Fig. 1d), enabling direct coupling with downstream analysis methods. Our initial experiments achieved DNA concentration factors of more than 100-fold and catalyzed follow-up efforts to determine the underlying mechanism and further increase concentration factors.

In order to identify the mechanisms driving MRT, we performed a series of preliminary experiments (Supplementary Note 1 and Supplementary Table 1) to test various experimental factors, including the influence of capillary surface charge and the effect of buffer composition of both the running and sample buffers. In parallel, we performed a numerical simulation using COMSOL Multiphysics® (see "Methods" section) to confirm the effect of these parameters and further probe the contributions of individual underlying forces (Fig. 2, Supplementary Fig. 1). Excellent qualitative agreement between the simulation (Fig. 1, bottom) and experiments (Fig. 1, top) were obtained, including the step of flow reversal and sample injection (Fig. 1d, h). From this, we eliminated what we felt was the most likely cause, the generation of a streaming potential (Supplementary Note 1), and instead determined that the development of an ion concentration gradient generates localized forces that propel DNA migration toward the capillary orifice.

The proposed mechanism is illustrated schematically in Fig. 2a along with supporting results from the simulation in Fig. 2b. Flow of the high ionic strength running buffer out of the capillary and into the low ionic strength reservoir buffer generates a concentration gradient of ions surrounding the capillary orifice. At a static interface, diffusion would quickly dissipate the concentration gradient; however, in our system, flow out of the capillary acts as a steady source of ions and the reservoir as a 3-dimensional and nearly infinitely large sink, allowing the

system to quickly equilibrate to a sustainable ion gradient. The ions expelled from the capillary can then diffuse down their concentration gradient and into the reservoir. The slower migration of cations relative to their anion counterparts generates an induced electric field (rainbow contour in Fig. 2b) that reaches a maximum <5 μm from the capillary orifice and decreases as it spans more than 50 μm into the reservoir (Supplementary Fig. 2). Remarkably, we find that this electric field generates quickly (under 5 s), persists for long time periods (at least 45 min), and can reach a maximum magnitude approaching 40 V/cm. This resulting electrophoretic force drives the negatively charged DNA molecules in the reservoir to migrate against the ion concentration gradient toward the capillary orifice (Fig. 2b, green arrows) to the region at which it is balanced by the hydrodynamic force from the capillary flow, causing the DNA to accumulate into a concentrated bolus. The hydrodynamic force acts to disrupt the concentrated DNA while the electrophoretic force acts as a restoring force. Interestingly, the highly localized nature of these forces means that they are not spatially uniform even within the bolus region, causing the DNA to recirculate within the bolus (Fig. 2b, red arrows). Flow out of the capillary into the large reservoir also increases the size of the buffer interface region, increasing the volume over which the DNA can concentrate and allowing for the generation of even higher concentration factors.

Solute concentration gradients in the absence of other external forces were first seen to effect migration of colloid particles as early as 1947[29]. This phenomenon, termed diffusiophoresis, has since been well studied both theoretically and experimentally within the colloidal research community[30–35], but only recently has been expanded to manipulate biological species including DNA[36–39], proteins[39–42] and cells[39, 43]. MRT invokes the same forces to establish an ion diffusion-generated electric field, but utilizes counterflow from a micro-orifice to generate a stable and highly localized ion gradient, increase ion flux, provide a counterbalancing force to DNA migration, and control the shape and location of the DNA focusing region. To our knowledge, diffusiophoresis alone has not been shown to generate the large

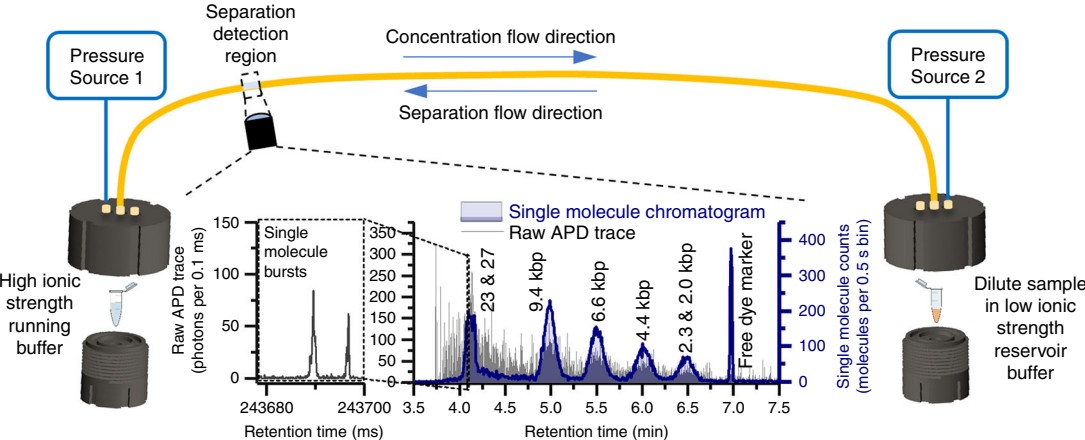

**Fig. 3** Coupled preconcentration, free solution hydrodynamic separation, and single-molecule detection platform. Each end of a long microcapillary is placed into its own individually controled pressure chamber to enable precise control over the direction and speed of fluid flow. Concentration occurs by applying positive pressure to the running buffer pressure chamber with Pressure Source 1, and sample injection and separation occur with positive pressure applied to the opposite end of the capillary with Pressure Source 2. The CICS observation volume is aligned with the detection region to enable detection of single DNA molecules (grey, background), which are counted to generate a single-molecule chromatogram (navy, shaded foreground). This example chromatogram was generated after performing 100× MRT preconcentration and free solution separation of HindIII digested λ DNA

concentration factors or high stability over time achieved with our flow-based method.

**Effects of flow parameters on concentration enhancement.** To quantify the concentration factors experimentally, we integrated MRT with SML-FSHS, a single-molecule separation platform that also operates in the absence of an applied electric field. The combined platform, illustrated schematically in Fig. 3, uses only pressure to control the direction, speed, and duration of flow during the sequential steps of MRT preconcentration, sample injection, and free solution hydrodynamic separation steps. Cylindrical illumination confocal spectroscopy (CICS) is used to individually detect each size-separated DNA molecule by its discrete fluorescent burst[6]. The single-molecule chromatogram generated by summing these single-molecule bursts enables quantification for each DNA fragment size by absolute number of molecules rather than arbitrary fluorescence units (see "Methods" section). This single-molecule analysis not only enhances SML-FSHS detection sensitivity, but it also improves quantification accuracy compared to bulk fluorescence intensity through direct counting and by minimizing the impact of fluorescence artifacts (Supplementary Fig. 3)[27, 28, 44].

We first characterized the effects of the counterflow rate (backpressure) on the DNA concentration enhancement by analyzing TOTO-3 stained HindIII digested λ DNA with 25 mM Tris-HCl as the running buffer and water as the low ionic strength reservoir buffer. In Fig. 4a, we report the concentration factor of each DNA fragment after 30 s of MRT concentration at a range of backpressures. The concentration factor increased from 0 to 50 psi. At 50 psi, a concentration enhancement >10-fold is achieved for all DNA fragments. This agreed with our initial observations, in which DNA concentrated in response to pressure-driven flow (Supplementary Movies 1 and 2). Above 50 psi, the concentration factor decreased with increasing flow rates. This observation suggests an optimal pressure where low flow rates enhance the bolus formation, but high flow rates begin to disrupt it. We saw this visually manifest in the following two forms: (1) the center of the bolus was blown out to form a donut-shaped concentration region, which grows with increasing backpressure (Supplementary Fig. 4 and Supplementary Movies 3–5) or, infrequently, (2) the concentration bolus became unstable and was dislodged from the capillary

(Supplementary Movie 6). Donut-shaped concentration was also observed in the simulation (Supplementary Fig. 5).

Next, we characterized the effect of concentration time at the optimal backpressure of 50 psi (Fig. 4b). The concentration factor increased with time for all DNA species. At 25 min, all DNA fragments were concentrated >2000-fold. Minor size-dependent concentration bias was seen after 25 min; the concentration factor of the largest molecules (27 and 23 kbp) was ~4-fold higher than that of the smallest (2.3 and 2.0 kbp). This small bias could be related to the variety of forces operating in MRT: purely electrophoretic techniques do not typically exhibit a substantial concentration size-bias for this DNA size range[45, 46], and diffusiophoretic migration of colloid particles can be size-dependent or -independent depending on the parameter space[35]. However, the bias is minimal compared to the overall $10^3$ concentration factors and could also be a result of fluorescence artifacts that cause undercounting of small DNA molecules (Supplementary Fig. 3).

Beyond enhancing sensitivity, MRT could also simplify SML-FSHS operation by eliminating the sample plug injection. Typically, sample plugs are generated by sequentially injecting running buffer, sample, and elution buffer. However, MRT generates a very small bolus of highly concentrated DNA at the capillary inlet that can be injected and separated without switching to elution buffer. The DNA sample itself is so dilute that it can act as an elution buffer. We demonstrated the feasibility of this single-step separation technique by preconcentrating the sample ~2000-fold prior to a continuous injection and separation. Minimal loss in resolution was seen compared to the traditional injection scheme (Supplementary Fig. 6).

**Effects of buffer ions on concentration enhancement.** The mechanistic resemblance to diffusiophoresis led us to investigate the potential contribution of a second chemiphoretic mechanism in which favorable interactions between positively charged cations and the negatively charged DNA molecules could also drive DNA migration (Fig. 2). Though not accounted for in our numerical simulation, the chemiphoretic component has been suggested to play a dominant role in diffusiophoresis in some circumstances[47].

To determine the relative contributions of electrophoresis and chemiphoresis, we designed a buffer system where we could hold

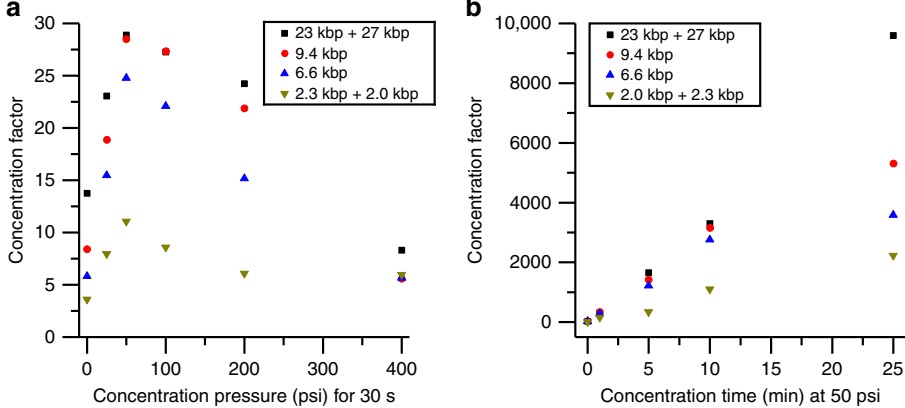

**Fig. 4** Effects of pressure and time on the concentration enhancement. **a** High concentration running buffer (25 mM Tris-HCl buffer) is pumped through the capillary (5 μm nominal inner diameter, 150 μm outer diameter, 120 cm total length) into the reservoir (λ DNA HindIII digest in DI water) for 30 s at varying pressures. The highest concentration factor occurs at 50 psi pressure. **b** Using the optimal 50 psi concentration pressure, concentration time is varied from 30 s to 25 min. The concentration factor increases with time for all DNA species

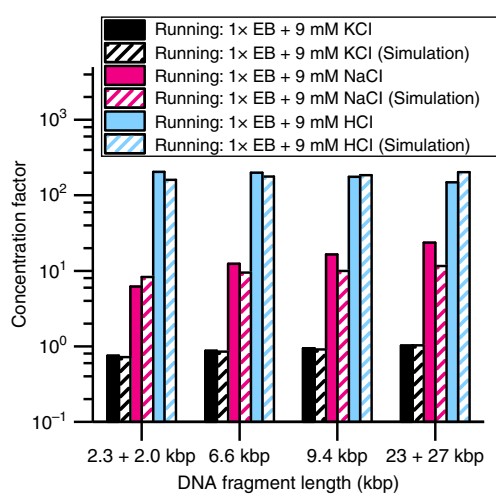

**Fig. 5** Effect of running buffer cation species on MRT concentration factor. The same buffering species are used in both the reservoir and running buffers: 1× EB (10 mM EACA and 40 mM Bis-Tris). Running buffer contains an additional 9 mM of the indicated species (KCl, NaCl, HCl). The concentration factor after 2 min of counterflow at 200 psi is plotted from experimental (solid) and simulated (striped) results. When KCl is added to the running buffer (black), the DNA does not concentrate above the starting concentration. When NaCl is added to the running buffer (pink), a 10-fold concentration effect is observed. The running buffer containing HCl (blue) results in the highest concentration factor—over 100-fold

the chemiphoretic component relatively constant while modulating the electrophoretic component. In these experiments, a low ionic strength base buffer was used in both the reservoir and running buffers (see "Methods" section). This "EB buffer" base consists of a weak acid (ε-aminocaproic acid, EACA) and a weak base (Bis-Tris), which titrate each other while remaining > 90% neutral and maintaining a high buffering capacity[48]. Three capillary running buffers were created with low, medium, and high expected electrophoretic components through the addition of 9 mM KCl, NaCl, and HCl, respectively. These species completely dissociate in solution and form an ion concentration gradient across the capillary orifice. With $Cl^-$ and EB concentrations held constant, these running buffers therefore allow us to probe the effect of cation diffusivity on the size of the induced electrophoretic field and the DNA concentration factor.

For the low electrophoretic force KCl buffer (Fig. 5), virtually no concentration enhancement was observed in either simulation (striped) or experiment (solid). Since $K^+$ and $Cl^-$ have very similar diffusivities, their diffusive fluxes are reported in the numerical simulation to be similar (Fig. 6a), minimizing the electrophoretic component (Fig. 6d). In this condition, any DNA concentration would be largely due to chemiphoresis. However, the low concentration factor for the KCl case suggests that chemiphoresis alone does not significantly induce DNA migration in our system.

For the medium electrophoretic force NaCl buffer, a DNA concentration factor of ~10-fold was seen in both the simulation and SML-FSHS experiment (Fig. 5). $Na^+$ has a lower diffusivity than $Cl^-$ ($K^+ \approx Cl^- > Na^+$). This results in a lower diffusive flux (Fig. 6b) and an induced electric field reaching 15 V/cm—more than 3-fold larger than that generated with KCl (Fig. 6e). These results suggest that only the electrophoretic component is the major contributor to MRT concentration factor.

For the high electrophoretic force HCl buffer, concentration factors greater than 100-fold were achieved (Fig. 5). $H^+$ has the highest diffusivity of all three cations ($H^+ >> K^+ > Na^+$). However, reactions with water and the buffering species (EACA and Bis-Tris) effectively scavenge the free $H^+$, yielding $H_3O^+$ and Bis-Tris$^+$. At pH 7.0, Bis-Tris$^+$ holds most of the positive charges and becomes the majority cation. The diffusive flux of Bis-Tris + in this buffer is ~10-fold larger than its flux in the KCl and NaCl cases, but still 3- and 4-fold smaller than the flux of $Na^+$ and $K^+$, respectively (Fig. 6a–c). The more than 4-fold difference in flux between Cl- and Bis-Tris + generates the largest induced electric field (Fig. 6f), and consequently the largest DNA concentration factors.

We then used the EB buffer system to examine the effect of the sample buffer composition. Figure 7 shows single-molecule separations performed on DNA in EB buffer with no MRT, DNA in DI water with MRT, and DNA in EB buffer with MRT. The DNA in DI water achieved over 1000-fold concentration enhancement after 12 min of preconcentration at 100 psi (Fig. 7b). In contrast, the DNA in EB buffer required 12 min of preconcentration at 400 psi to achieve the same 1000-fold enhancement (Fig. 7c). We hypothesize that the concentration gradient of EB buffer into a water reservoir results in a higher diffusive flux of the Bis-Tris$^+$ ions, resulting in a lower optimal flow rate. Simultaneously, the higher conductivity of the EB reservoir compared to water could also affect the magnitude and span of the electric field, which is compensated with a higher flow

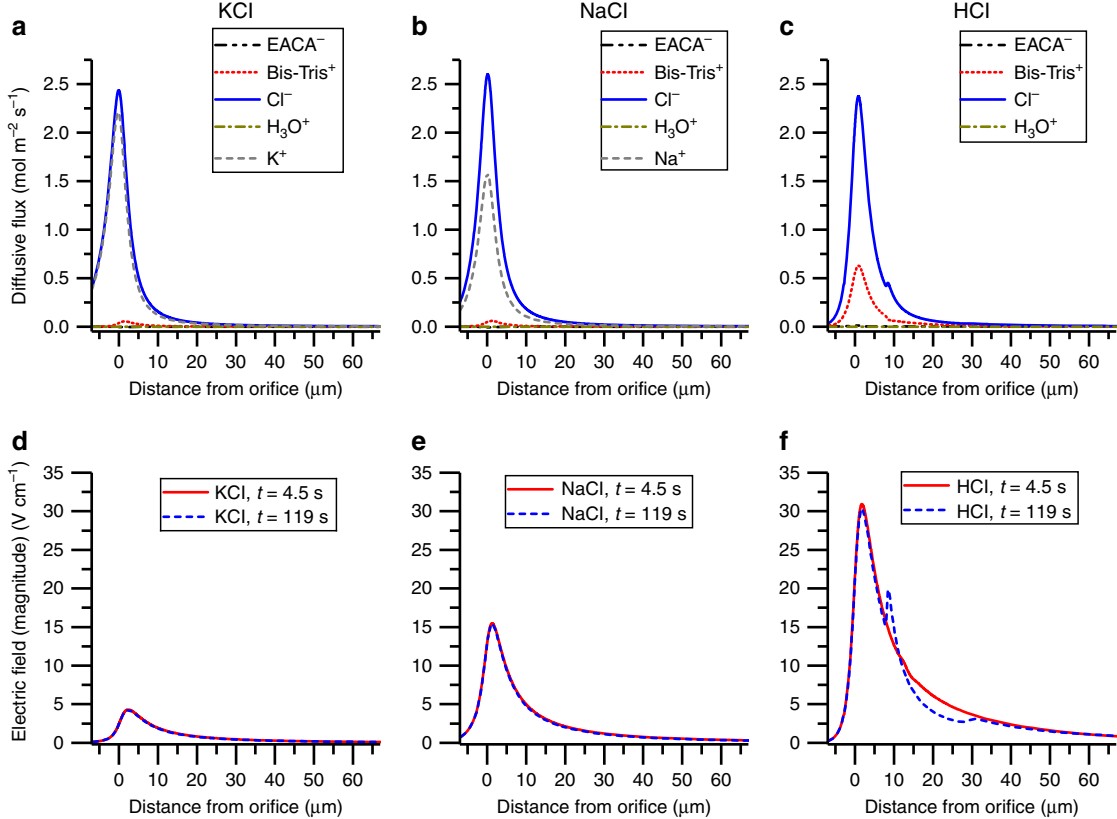

**Fig. 6** Centerline ion diffusive fluxes and magnitude of the induced electric field in KCl, HCl, and NaCl simulations. The simulation conditions are the same as Fig. 5: both reservoir and running buffers contain 1X EB; running buffer contains an additional 9 mM of KCl (left), NaCl (middle), or HCl (right). Diffusive fluxes of ions at 119 s time point in **a** KCl, **b** NaCl, and **c** HCl are plotted along the centerline of the capillary as a function of distance into the reservoir. The capillary orifice is located at $x = 0$. $Cl^-$ is the majority anion carrier in all cases (blue, solid). In KCl **a**, the diffusive flux of $K^+$ (grey, dashed) is almost as high as $Cl^-$, and the rest of the ions are negligible. In **b**, $Na^+$ (gray, dashed) has the second largest flux, with the other ions negligible. In HCl **c**, Bis-Tris$^+$ (red, dotted) becomes the majority cation, though its flux is smaller than both $Na^+$ and $K^+$. The magnitude of the induced electric field generated by these ion fluxes is plotted at 4.5 s (red, solid) and 119 s (blue, dashed) time points in **d** KCl, **e** NaCl, and **f** HCl. The largest field is generated in HCl and extends tens of microns into the sample reservoir. NaCl generates the second largest field, and KCl generates the smallest. The field magnitude and shape are highly stable with time for all 3 conditions, except where the concentrated DNA bolus somewhat distorts the field in HCl at 119 s

rate. A comparison of the separation resolution between the three chromatograms in Fig. 7 shows <7% CV, verifying that MRT preconcentration does not negatively affect the separation efficiency. Additional control experiments (Supplementary Figs. 8 and 9) validate that the high concentration factors require both counterflow and the running and reservoir buffer mismatch.

Finally, we attempted to maximize the DNA concentration factor by integrating all of the previously optimized conditions (Fig. 8). By switching from the initial Tris-HCl buffer to an optimized EB-HCl buffer, we observed a 2-fold increase in the optimal counterflow rate as well as a 2–3-fold increase in the concentration factors within the same concentration time. The EB-HCl buffer generates a larger electrophoretic field than the Tris-HCl buffer, which allows the DNA migration to overcome higher counterflow velocities. We hypothesize two mechanisms that could lead to this result. First, faster DNA migration due to larger electrophoretic fields in the EB-HCl buffer enables the DNA migration to overcome higher counter-flow velocities. Second, the higher counterflow rate expands the buffer interface and acts to attract DNA from a larger sample volume. Then, by increasing concentration time from 10 to 45 min, we were able to achieve concentration factors exceeding 10,000-fold (Fig. 8). While the results from the numerical simulation (striped bars) in Fig. 8 show higher deviation from experiment (solid bars) than those in Fig. 5, they do present the same trends as the experimental results with the same

experimental conditions achieving the highest concentration factors. The >$10^4$ concentration factors achieved experimentally increased the effective sample screening volume from ~45 pL (injected volume) to ~0.5 μL (preconcentrated volume containing the same number of DNA molecules), enabling SML-FSHS analysis from a starting sample concentration of only 150 aM (<100 copies per μL). On an average, we counted ~500 molecules per fragment size, well above a projected limit of detection (S/N = 3) of ~40–120 molecules, which would correspond to a sample concentration of 10–50 aM. In our previous work without preconcentration, we projected SML-FSHS limit of detection to be ~3.5 pM, which was a 2–3 orders of magnitude improvement over existing methods[27]. The addition of MRT improves SML-FSHS sensitivity by an additional 5–6 orders of magnitude, making it among the most sensitive amplification-free, size-based analytical methods.

## Discussion

In this work, we demonstrated a simple, unexpected method to concentrate DNA at a discontinuous buffer interface without an applied electric field. In this technique, based on the migratory phenomenon that we call MRT, DNA in a low ionic strength buffer reservoir migrates toward the outlet of a capillary dispel-ling a high ionic strength buffer driven by a pressure gradient. Complementary numerical and experimental approaches were

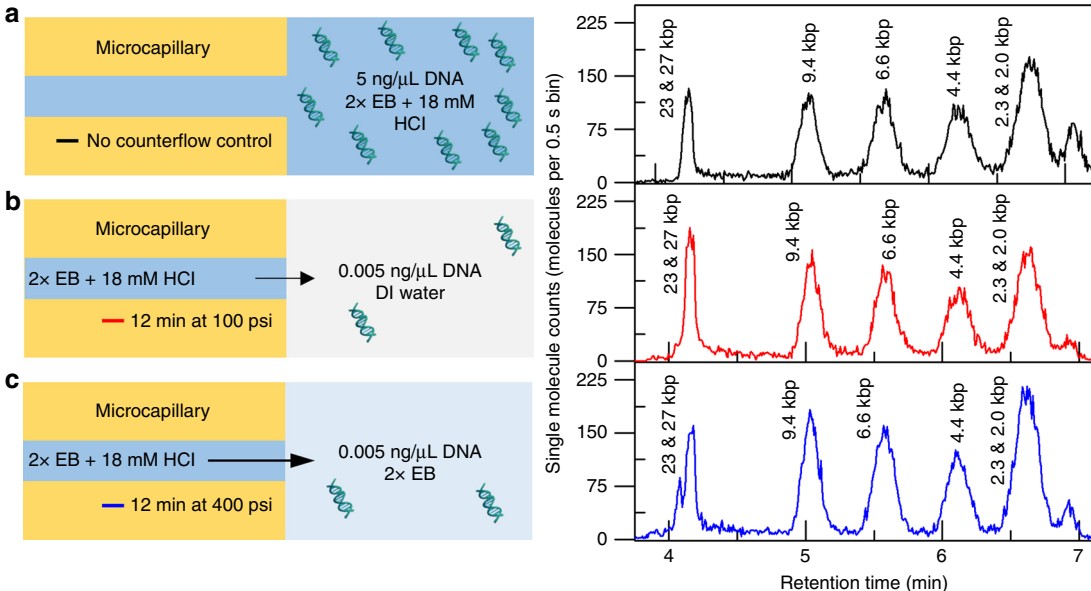

**Fig. 7** Effect of reservoir buffer solution on MRT preconcentration. A schematic describing the experimental preconcentration conditions is shown on the left, and the single-molecule chromatogram after separation is shown on the right. In all conditions, 2× EB + 18 mM HCl was used as the running buffer. **a** Control separation of λ DNA *Hind*III digest without preconcentration or a buffer mismatch: the reservoir buffer is the same as the running buffer and no counterflow is applied. **b** Counterflow at 100 psi for 12 min into reservoir buffer containing *Hind*III digested λ DNA in DI water. The DNA, originally suspended at 0.005 ng/μL concentrates 1000-fold prior to separation. **c** A similar 1000-fold concentration factor is achieved with the reservoir buffer solutions containing DNA prepared in 2× EB by flowing at a 4-fold higher flow rate (400 psi for 12 min)

used to develop the proposed underlying mechanism, which involves an ion gradient-induced electric field (as in diffusiophoresis) and a counteracting microfluidic flow to both generate the ion gradient and focus the DNA. Though diffusiophoresis does enable species-specific manipulation over macroscopic length scales in a non-contact manner, we surmise that the slow adoption of diffusiophoresis is due to the challenge of establishing a stable solute concentration gradient for extended time periods[32]. To overcome this challenge, researchers have used semi-permeable membranes[31, 40], co-flow microfluidic devices[49], reactive surfaces[50–53], thermal gradients[43, 54, 55], and micro-devices with flow channel reservoirs[35, 36, 43, 56]. In our technique, pressure-driven flow of a high ionic strength buffer out of the capillary and into a low ionic strength buffer reservoir generates a highly local but stable ion concentration gradient and induced electric field that is able to scavenge DNA from the reservoir for extended time periods. The pressure-driven flow also provides an additional hydrodynamic force acting to balance the induced electrophoretic migration of the DNA. DNA then accumulates into a highly concentrated region surrounding the capillary orifice.

Through optimization of the pressure, time, and buffer conditions, we have demonstrated concentration factors approaching 5 orders of magnitude, comparable to traditional DNA preconcentration methods such as FASS and ITP. In both FASS and ITP, a buffer mismatch is used to manipulate an externally applied electric field to cause DNA to preferentially accumulate at the interface[57, 58]. DNA analysis with FASS-CE, ITP-CE, and related electrokinetic platforms can boast limits of detection in the range of low pM to tens of fM[13, 59, 60]. In our technique, the electric field is generated internally via pressure-driven flow and diffusion, eliminating the need for an external input. The coupled MRT-SML-FSHS platform has a limit of detection approaching tens of aM, a 3–5 orders of magnitude improvement over traditional electrokinetic platforms.

Moreover, we anticipate that further optimization could be done to obtain even higher concentration factors. Though the sustainability of MRT preconcentration with time has not yet been explicitly determined, we hypothesize that parameters such as the shape and volume of the sample reservoir and micro-capillary could influence both the MRT duration limit and concentration rate. Second, the use of a higher ion concentration in the running buffer to increase the solute concentration gradient (i.e., the diffusive driving force) could also enable increased concentration factors. We did not test higher ionic strength buffers in this study because of the staining requirements of the fluorescent DNA staining dye, but we hypothesize that this may enhance the concentration scheme in the following two ways: (1) an increase in the volume of the reservoir spanned by the solute concentration gradient and (2) an increase in the DNA migration speed. Similarly, a combination of mono- and multi-valent salts may be able to boost the induced electric field and increase the speed or concentration capability of the MRT scheme. Finally, we propose that inducing gentle mixing within the sample solution could increase the achievable concentration factors in a similar manner to what has been demonstrated for electrokinetic preconcentration[61].

One important consideration for implementing MRT to concentrate DNA from biological samples is that many biofluids (e.g., serum) have relatively high salt concentrations. We suggest that this concentration method could be adapted to work with these samples by reversing the mode of operation: instead of adding the DNA sample to the reservoir, it could be introduced in the high ionic strength running buffer that flows through the capillary. DNA molecules exiting the capillary would be trapped at the capillary orifice and accumulate with continued flow over time. We did perform concentration in this mode to validate the concept; however, concentration occurred more slowly due to the low volumetric flow rate that limited DNA flux to the interface. Using a larger diameter capillary could help to overcome this volumetric flow rate limitation, although the relationship between capillary diameter, flow rate, and MRT concentration has not yet been tested.

Given the robustness and simplicity of MRT, we foresee utility in a wide range of applications such as in microfluidic devices,

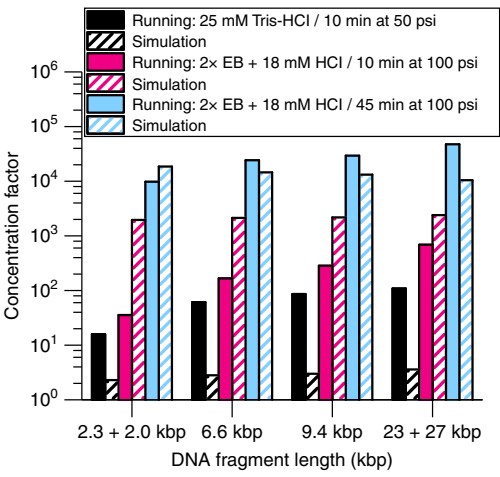

**Fig. 8** Concentration factors increase with optimized buffer solutes and longer concentration times. Experimental (solid) and simulated (striped) results compare concentration factors with 25 mM Tris-HCl running buffer (black) with 100 mM EBH-18 running buffer at 10 min (pink) and 45 min (blue) at each buffer's optimal flow rate. Simulated (striped) and experimental (solid) results show the same trend of increasing concentration factors with increased concentration time and mobility difference between the majority cation and anion in the running buffer. The 2× EB + 18 mM HCl running buffer has a greater mobility difference between the majority cation and anion and generates higher concentration factors than the 25 mM Tris-HCl running buffer over the same counterflow time. Further increase of the counterflow time allows more DNA to accumulate in the injection region for higher concentration factors. Concentration factors over 10,000-fold were achieved experimentally with 45 min of preconcentration counterflow

point-of-care sample handling, and biomolecular assays, both as a stand-alone technology and coupled to subsequent molecular analysis operations. The integration of MRT with SML-FSHS represents a truly electrode-free platform for highly sensitive and quantitative DNA separation and sizing.

## Methods

**Mathematical model**. In order to test the hypothesis that focusing of dsDNA is caused by a diffusion-induced electric field, a mathematical model was prepared, including a 120-centimeter long capillary with a 5 micron-diameter lumen and a $\zeta$-potential in the range, $-60\,\text{mV} < \zeta < 0\,\text{mV}$, which opens abruptly into a 5 μL reservoir containing a very dilute solution, $< 5 \times 10^{-5}\,\mu\text{g}/\mu\text{L}$, of dsDNA. Both the capillary and the sample reservoir are initially filled with a low-conductivity sample buffer and, at $t = 0$, a higher conductivity solution is pumped through the capillary toward the reservoir.

COMSOL Multiphysics® is a commercial finite element software program that allows the simultaneous simulation of multiple, coupled transient transport equations in a geometry fabricated in 0-, 1-, 2- or 3-dimensions. We started building our model in COMSOL by choosing a 2D-axisymmetric coordinate system, then assembling the geometry to which we would apply our model equations (Supplementary Fig. 10), adding information regarding the properties of the materials in our model and then setting the model equations, boundary conditions and initial conditions.

In this model, an applied pressure of 200 psi across a 120 cm capillary with a 5 μm lumen produces a flow velocity <0.03 cm/s, which yields a lumen Reynolds number below 0.001 for water at 25 °C. Therefore, a vortex-free laminar flow can be assumed so we can use the incompressible Navier–Stokes equations together with the equation of continuity,

$$\rho \frac{\partial \mathbf{v}}{\partial t} + \rho \mathbf{v} \cdot \nabla \mathbf{v} = -\nabla p + \nabla \cdot \mu_{\text{eff}} \nabla \mathbf{v} \tag{1}$$

$$\nabla \cdot \mathbf{v} = 0 \tag{2}$$

to model both the flow, $\mathbf{v}$, from the capillary lumen into the sample reservoir as well as the reverse flow as DNA is drawn into the lumen. Note that since this is a closed flow system, the gravitational body force term has been subsumed into the augmented pressure, $p$, and natural convection is neglected. The effective viscosity

term takes into account the impact of the concentrated DNA on the viscosity of the electrolyte solution and $\rho$ is the density of the solution.

The Nernst–Planck constitutive equation for the mass flux is used in the equation of conservation of mass,

$$\frac{\partial c_i}{\partial t} + \mathbf{v} \cdot \nabla c_i + \nabla \cdot (z_i \omega_i c_i F \mathbf{E}) = \nabla \cdot D_{\text{eff},i} \nabla c_i \tag{3}$$

to describe the behavior in solution of the electrolyte and buffer ions as well as the highly-charged, double-stranded DNA, and includes the effects of diffusion, convection and electrophoretic migration on strongly-coupled ionic transport. Here $c_i$ is the concentration of the $i^{\text{th}}$ species, $\omega$ is its absolute electrophoretic mobility, $z$ is its valence, $\mathbf{E}$ is the electric field and $F$ is Faraday's constant.

The Poisson equation,

$$\nabla \cdot \varepsilon_r \varepsilon_0 \nabla \mathbf{E} = \sum_{all\,species} z_i F c_i \tag{4}$$

is used to calculate that portion of the electric field which arises as a result of the $\zeta$- potential near the interior and exterior surfaces of the capillary. Although the induced electric field and double-layer electric field are parts of a common electric field, in practice, we find it simpler and computationally more stable to calculate them separately and superimpose these two fields than to calculate them together.

The no-slip condition is applied on all solid surfaces, all glass surfaces are assumed to have a constant $\zeta$-potential, to be electrically insulating and to be impermeable to the ions and to the flow. Finally, a point at the far end of the reservoir is set to ground and the upstream end of the capillary is assumed to be insulated so that the net current across the capillary is zero.

Initially, there is no flow anywhere, the low-conductivity dsDNA-containing sample fluid fills the entire reservoir and penetrates about 120 microns into the capillary, while the rest of the capillary is filled with the high-conductivity eluent buffer. The applied pressure driving the flow from the capillary into the sample reservoir starts after 2 s and ramps from zero to full pressure in < 0.25 s. At the end of the focusing period, the applied pressure is ramped down to zero and held at zero for 2 s before the reverse pressure is applied.

**Reagent and buffer preparation**. A wide range of reservoir and running buffers were used in this work. The properties of these buffers, their shorthand abbreviations, their MRT buffer role (reservoir buffer or running buffer), and the corresponding figure numbers are listed in Supplementary Table 2. TE buffer (10 mM Tris-HCl, 1 mM EDTA, pH 8.0) was purchased from Ambion. Sodium Chloride (NaCl) 5 M solution was purchased from Amresco. Potassium chloride (KCl) 8 M solution was purchased from Sigma-Aldrich. Hydrochloric acid (HCl, 35%) was purchased from Sigma-Aldrich. All remaining buffer reagents (Bis-Tris, EACA, HEPES, and Tris) were purchased in dry form from Sigma-Aldrich and dissolved in filtered deionized water. Polyvinylpyrrolidone (molecular weight 360,000, Sigma-Aldrich) was prepared in TE buffer at 2% w/v.

**Sample preparation**. For fluorescent microscope experiments, HindIII digested λ DNA (New England Biolabs, Inc.) was diluted to 10 ng/μL and stained with 40× or 100× dilution of PicoGreen (Thermo Fisher Scientific, Inc.) in 1 mM HEPES-Tris buffer. For SML-FSHS experiments, HindIII digested λ DNA was diluted to 5 ng/μL and stained with 1 μM TOTO-3 Iodide (Thermo Fisher Scientific, Inc.) in either water or buffer. The samples were either used at this concentration or further diluted into the sample buffer before each experiment. Alexa 647-NHS ester "free dye" was used as a control to normalize retention times between experiments. It was stored at a stock concentration of 1 μM in water and spiked into each sample at 500 or 100 pM concentration.

**Fluorescent micrograph and video collection**. All fluorescent images were taken using an upright epifluorescent microscope (BX51, Olympus) with a 10× air objective (UPlanFl, Olympus). A 470 nm LED (ThorLabs) provided fluorescent excitation light. Fluorescent emission was collected with a CCD camera (Regita Exi, QCapture) with 100 or 500 ms exposure. Images were collected once per second using the MicroManager plugin for ImageJ.[62] ImageJ was used to assemble the images into movies at 10× speed.

**Capillary preparation and separation protocol**. A 5 μm nominal inner diameter fused silica capillary (Polymicro Technologies, Molex) was cut to a total length of 120 cm. A short section of the polyimide coating was burned from the capillary to form a viewing window 90 cm from the inlet (capillary effective length) that served as the detection window for the size-based hydrodynamic separations following MRT preconcentration. The inlet and outlet of the capillary were housed in their own pressure injection chambers which contained either 100 μL (inlet) or 200 μL (outlet) of reservoir or running buffer.

Flow was driven through the capillary using an argon pressure source, which was regulated at each end of the capillary by separate LabVIEW-controlled precision pressure regulators. The capillary was first filled with the running buffer (see Supplementary Table 2; e.g., 25 mM Tris-HCl, 2× EB, etc.) from the injection side. Pressure was stopped and the tube of injection buffer was replaced with a tube

containing the stained DNA sample in the reservoir buffer (e.g., water, EB buffer, etc.). Then pressure was applied to the outlet pressure chamber to force counterflow from the capillary into the reservoir at a specified pressure and time. Immediately after the counterflow pressure was vented, 50 psi pressure was applied to the capillary inlet for 10 s to inject a concentrated sample plug. After the injection pressure was vented, the sample tube in the inlet chamber was replaced with 100 μL of running buffer. The inlet chamber was pressurized to 450 psi for the duration of the separation. Retention time is counted from the application of the separation pressure. Flow rates were calculated from the retention time of the free alexa dye marker and the effective capillary length. The capillary was always rinsed with running buffer at least once between each separation for at least 11 min (~1.5 column volumes) before concentrating and separating a new sample.

**Data collection and analysis**. Single-molecule detection was performed with the CICS platform described previously.[28] Briefly, the capillary viewing window was aligned within the platform's optical observation volume. A 640 nm laser diode at 8 mW power provided fluorescent excitation, and photon counts were collected from the avalanche photodiode detector in 0.1 ms bins using a custom LabVIEW program. Fluorescent bursts from single molecules were detected and counted using thresholding analysis. Single-molecule bursts were counted in 0.5 or 1 s bins and plotted as a chromatogram. Peak areas were obtained by fitting the chromatograms to a series of Gaussian peaks. The area of each peak was used to calculate the total number of molecules $N$ present of each separated fragment size.

The concentration factor $C_F$ was calculated for each experiment by normalizing the results after concentration to a no-concentration control separation performed in the same running buffer. The calculation was performed individually for each fragment size using Eq. 5 below,

$$C_F = D_F \frac{N}{N_C}, \tag{5}$$

where $N$ and $N_C$ refer to the number of counted molecules of a given fragment size after concentration and in the no-concentration control, respectively. $D_F$ is the dilution factor of the starting DNA sample concentration from the sample used for the control separation. SML-FSHS quantification repeatability and calculation of the concentration factor for MRT-SML-FSHS was found to be within 50% for experiments done on the same day, as shown in Supplementary Fig. 11.

It should be noted that the rectangular aperture used to eliminate out-of-plane fluorescence limited the dimensions of the observation region to a $4 \times 1$ μm slit along the capillary length. Molecules near the wall of the 5 μm diameter capillary could have passed outside the observation volume without being detected, limiting the mass detection efficiency of single-molecule counting. However, this should not impact the ratiometric calculation the concentration factor because the fraction of the number of counted molecules to the number of actual molecules for any given fragment size should be the same for all separations.

**Code availability**. The code written to perform the numerical simulations and to collect and analyze the single-molecule data is available from the authors upon reasonable request.

**Data availability**. The data that support the findings of this study are available from the authors upon reasonable request.

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

## Acknowledgements

S.M.F. and T-H.W. were supported by supported by grants from the NIH (R01CA155305, R21CA186809 and R01AI117032). J.M.B. and K.J.L were supported by grants from the NIH (2R44GM103356 and 1R43GM103356). C.F.I. was supported by the Paul M. Hohenschuh Distinguished Professorship in Chemical Engineering at WSU.

## Author contributions

S.M.F. and J.M.B.: Performed experiments and analyzed the data. C.F.I.: Performed the numerical simulations. All authors discussed the results and contributed to writing and revising the manuscript.
