## [Peer Review File · Nature Communications]

Reviewers' comments:

Reviewer #1 (Remarks to the Author):

this manuscript describes interesting experiments in which DNA (and presumably other molecules) can be pre-concentrated by many orders of magnitude using simple pressure-driven flow of the sort that one would employ in normal microfluidic operations. Typically, pre concentration is achieved using electric fields (field-amplified sample stacking, isotachopheresis, etc.) and requires a more involved operation. The trick here is to inject a solution with higher ionic strength than the reservoir. The authors term this "molecular rheotaxis", and rightly describe its physical underpinning in terms of diffusiophoresis, known from the colloidal context.

I think this is a very clever experiment, whose results are very compelling, and which I do expect will both enable new applications and also help researchers understand unexpected behavior that they might see under similar circumstances. The article is clearly written (I appreciate the narrative account of its accidental discovery — a refreshingly honest account of how science often proceeds) — and the results are compelling. I recommend publication in Nature communications.

My only suggestions involve how these results are related to diffusiophoresis. A few thoughts here — first, I think that MRT really is a direct manifestation of diffusiophoresis, rather than a phenomenon "related to diffusiophoresis". The way in which the electrolyte gradients are established here are different from how those gradients were established in classical studies of diffusiophoresis, but nonetheless the core physical principle here is diffusiophoresis, and it would take nothing away from the achievements here to describe MRT as a new way to generate gradients that drive DP...

Second — whether "chemi-phoresis" or "electro-diffusiophoresis" is more important in the colloidal literature is a subtle question, and depends on the zeta potential of the colloids (or polymers) as well as the difference between ion mobilities. So it is incorrect to describe the literature with statements that make it sound like colloidal diffusiophoresis is dominated by one or the other, or blanket statements about size-dependence etc. The theory is well-developed for colloids, and predicts different results in different parts of parameter space. I remark on this because my read of the paper felt like it was trying to establish a difference between diffusiophoresis and MRT, which may not have been the intention, and moreover is not really accurate, and is certainly not necessary.

In all, though, I thoroughly enjoyed the work and recommend publication.

Reviewer #2 (Remarks to the Author):

Friedrich et al. report an interesting phenomenon of retrotransport of DNA towards an orifice from which a flow originates, resulting in accumulation and concentration of the sample close to the orifice, allowing the later injection of the DNA in this orifice by flow reversal. They characterize the phenomenon experimentally and show that numerical simulations account fully for the observed phenomenon, which they suggest could be used in microfluidic devices for sample concentration. I found the manuscript well written and solidly documented, if a bit complex, and recommend publication as is, with the inclusion of comments on a few remarks:

1. Onset of the flow (Line 117), quoted to be <5 s: is this reproduced by simulations? What governs this time scale? Can this be reduced? Similar question about the sustainability of the flow: is there a limit to its duration?
2. Larger diameter capillary (Line 312): what is the influence of capillary dimensions on this phenomenon?
3. What is/could be the influence of the DNA state (double-stranded versus single-stranded)? Could this technique be used for other molecules (proteins, charged polymers, etc.)?

Minor comments:

1. There are a lot of undefined abbreviations, which may be obvious for some but not all readers. Please define when first used. Some are listed below.

Line 48: define CE (capillary electrophoresis?)

Line 52: define EOF (electro-osmotic flow?)

Line 62: define HPLC, HDC (high-pressure liquid chromatography? Hydrodynamic chromatography?)

Line 74: what are "drag tags"?

Line 193: define EB

Line 195: define EACA, Bis-Tris

2. Line 217: where does 2000-fold come from? Fig. 5 shows 200.

3. Line 443, 446: "The data/code that support the findings of this study are available from the authors upon reasonable request". What is a "reasonable request"? In my experience, either the data is shared (that is deposited and described on a public online site such as Figshare) or not. In the former case, the benefit is that the data is granted a DOI, which can then be used for traceability. In the latter case, opening the door to conditional individual requests will rapidly become bothersome to answer individual requests (not mentioning data loss, first author moving to a new position and unable to help, etc). The same goes for code, which can be easily deposited on Github, Bitbucket or similar sites (rather than institutional sites which usually do a poor job of preserving permanent links). Note that I am not asking for either data or code, because I have no time to spend digging into either now, but I would hate to be told that some of my requests are unreasonable if I did.

4. There are a few typos, e.g. Line 291: "an limit of detection"

Reviewer #3 (Remarks to the Author):

The manuscript "Molecular Rheotaxis: Inducing DNA Migration and Concentration Against a Pressure-Driven Flow" by Friedrich et al, reports on a microfluidic preconcentration procedure for DNA, that is based on the well-known phenomenon, rheotaxis.

As mentioned, rheotaxis is a known concept (relating to organisms and particles, etc.), so the statement in the abstract that says 'This unexpected and counterintuitive migratory...' and in the discussion ('..new, unexpected method..' are redundant. Why would it be surprising? Extraction of analytes under counter-current flow is also known.

If the authors wish this to attract more attention for those outside the community, perhaps they should decide on a more accessible term, other than molecular rheotaxis, for the procedure. Those interested in preconcentrating, say aqueous contaminants, in microfluidic systems, may find potential use of the approach.

Generally, the work is interesting, it being about sample preconcentration in a microfluidic device, albeit only a single type of analyte, and could find applicability by those in these fields. One of the bugbears of microfluidic systems is that sample preparation (including cleanup, and enrichment (preconcentration)) is often still performed outside of the system, and needs additional efforts (involving in-house setups like those indicated by the authors in the introduction). Here, a potential inline approach (something that is integrated within the microfluidic system) is reported, that is seemingly easy to implement, may pave the way for truly practical use of micro-total analytical systems for real-world applications, although the broader appeal is unknown since the present work deals with DNA only (and it is not clear if the procedure works with mixtures of multiple components). It is also unclear if smaller molecules will be subjected to the same phenomenon.

Field-amplified sample stacking and isotachophoretic approaches are compared with the present one in terms of the limit of detection for DNA; one would have expected a more comprehensive

evaluation these and other existing procedures.

Generally, this work is publishable.

Minor comments:

The abstract should provide more quantitative data and information.

Is it necessary to present results and data at the end of the introduction?

Why is the capillary rinsed for 'at least 11 min?' How is this value arrived at?

Error bars are missing from the relevant plots.

Additional comments:

In several instances, abbreviations are repeatedly defined.

Check ref. 52: Author's name is completely capitalized; ref. 24, 22: Check the reporting of the reference; refs. 5, 10, 26, 29, 35, 36, 39, 40, 43, 55, 56: Check reporting format of respective journal titles.

General note to reviewers:

We thank each of the reviewers for their time, thorough review, and thoughtful comments. We have attempted to address the all of the specific and general comments in the revised manuscript. The major modifications include 1) incorporating additional results in the supplementary information, 2) revising the text to enhance correctness and clarity of difficult concepts, 3) adding references, and 4) providing additional experimental and analytical details. Below, please find our responses to each comment in blue and noted specific revisions to the manuscript text in red.

Reviewer #1 (Remarks to the Author):

this manuscript describes interesting experiments in which DNA (and presumably other molecules) can be pre-concentrated by many orders of magnitude using simple pressure-driven flow of the sort that one would employ in normal microfluidic operations. Typically, pre concentration is achieved using electric fields (field-amplified sample stacking, isotachopheresis, etc.) and requires a more involved operation. The trick here is to inject a solution with higher ionic strength than the reservoir. The authors term this "molecular rheotaxis", and rightly describe its physical underpinning in terms of diffusiophoresis, known from the colloidal context.

I think this is a very clever experiment, whose results are very compelling, and which I do expect will both enable new applications and also help researchers understand unexpected behavior that they might see under similar circumstances. The article is clearly written (I appreciate the narrative account of its accidental discovery — a refreshingly honest account of how science often proceeds) — and the results are compelling. I recommend publication in Nature communications.

My only suggestions involve how these results are related to diffusiophoresis. A few thoughts here — first, I think that MRT really is a direct manifestation of diffusiophoresis, rather than a phenomenon "related to diffusiophoresis". The way in which the electrolyte gradients are established here are different from how those gradients were established in classical studies of diffusiophoresis, but nonetheless the core physical principle here is diffusiophoresis, and it would take nothing away from the achievements here to describe MRT as a new way to generate gradients that drive DP...

We thank the reviewer for their insightful comments. We do agree that the physics underpinning DP are the same as those driving the counter-migration response in MRT. We also agree that one of the features distinguishing MRT from conventional DP is the way flow out of a microcapillary is used to generate the ion concentration gradient. However, the flow also serves an additional role within the focusing region by serving as an opposing force to the migrating DNA species. In fact, this flow produces the recirculating pattern within the bolus and influences the sampling of the reservoir region (DNA tends to enter the bolus from the capillary walls because of the streamline profile). We wanted to use nomenclature that both drew attention to the flow stimulus and effectively described the phenomenon's functional effects, which led us to MRT. Importantly, we did not intend to minimize the role of DP and we regret our choice of diction here. We have therefore revised the text to more clearly describe the relationship between MRT and DP as phenomenon that share a common basis:

Page 4: MRT invokes the same forces to establish an ion diffusion-generated electric field, but utilizes counterflow from a micro-orifice to generate a stable and highly-localized ion gradient, increase ion flux, provide a counterbalancing force to DNA migration, and control the shape and location of the DNA focusing region.

Page 7: Complementary numerical and experimental approaches were used to develop the proposed underlying mechanism, which involves an ion gradient-induced electric field (as in diffusiophoresis) and a counteracting microfluidic flow to both generate the ion gradient and focus the DNA.

Second — whether "chemi-phoresis" or "electro-diffusiophoresis" is more important in the colloidal literature is a subtle question, and depends on the zeta potential of the colloids (or polymers) as well as the difference between ion mobilities. So it is incorrect to describe the literature with statements that make it sound like colloidal diffusiophoresis is dominated by one or the other, or blanket statements about size-dependence etc. The theory is well-developed for colloids, and predicts different results in different parts of parameter space. I remark on this because my read of the paper felt like it was trying to establish a difference between diffusiophoresis and MRT, which may not have been the intention, and moreover is not really accurate, and is certainly not necessary.

We thank the reviewer for pointing out our oversimplification in the way we described these works. We have revised our text to make it clear that these should not be taken as blanket statements, and were meant to demonstrate the importance of characterizing these aspects within our system. The revisions to the text are listed below:

We have removed the statement on page 6: “This is in contrast to studies of diffusiophoresis of colloids that have found that the chemiphoretic component dominated the electrophoretic component.”

We have edited the statement on page 5 to read: This small bias could be related to the variety of forces operating in MRT: purely electrophoretic techniques do not typically exhibit a substantial concentration size-bias for this DNA size range⁴⁵⁻⁴⁶, and diffusiophoretic migration of colloid particles can be size-dependent or -independent depending on the parameter space³⁵.

In all, though, I thoroughly enjoyed the work and recommend publication.

Reviewer #2 (Remarks to the Author):

Friedrich et al. report an interesting phenomenon of retrotransport of DNA towards an orifice from which a flow originates, resulting in accumulation and concentration of the sample close to the orifice, allowing the later injection of the DNA in this orifice by flow reversal. They characterize the phenomenon experimentally and show that numerical simulations account fully for the observed phenomenon, which they suggest could be used in microfluidic devices for sample concentration. I found the manuscript well written and solidly documented, if a bit complex, and recommend publication as is, with the inclusion of comments on a few remarks:

1. Onset of the flow (Line 117), quoted to be <5 s: is this reproduced by simulations? What governs this time scale? Can this be reduced? Similar question about the sustainability of the flow: is there a limit to its duration?

The < 5 s timescale of the onset of MRT concentration was determined through analysis of the simulation results. We observed minimal changes in the shape of the electric field between early time points (4.5 s) and late time points (2691 s), indicating to us that (1) the flow and field were very responsive to the imposed pressure gradient and (2) both were also highly stable with time. In light of the reviewer's question, we used the simulation to further probe the evolution to steady state of both the flow velocity and the corresponding induced electric field as a function of time. We have added this analysis to Supplementary Fig. 2 in panels (c) and (d). We have also included the following discussion into the Supplementary Fig. 2 caption:

To further probe the degree of the stability and responsivity of MRT, domain-averaged magnitude values of the velocity (solid) and electric field (dashed) are plotted over (c) the full concentration period and (d) the first 5 seconds of concentration. These values were computed over 3 domain sizes: 25x the capillary radius (black), 50x the capillary radius (red), and the full reservoir domain (blue), and normalized to the values at the final time point of applied pressure. Over the full reservoir domain, it takes ~50 s for the fluid flow to stabilize, and the electric field continues to change with time. This is likely due to the low volumetric flow rate (on the order of pL/s) that limits the speed at which the ions can span the full 5 μL reservoir region. However, the bolus concentration region is situated closely to the capillary orifice (< 25 μm, (b) inset) and is thus contained within a much smaller domain where both the velocity and the electric field stabilize much more quickly. Within the 62.5 μm domain (black), the velocity curve closely follows the imposed pressure ramp rate, and the electric field reaches its maximum with only ~1s lag. The remainder of the reservoir serves as the sink, allowing the field within the bolus region to remain relatively stable.

Once the flow reached steady state, it is highly stable and can be left to run indefinitely (as shown in new Supplementary Fig. 2c). However, the sustainability of MRT concentration has not been fully determined. We are very interested in characterizing the concentration time limit and plan to address this question in future research. We have included a brief statement addressing this question in the discussion section:

Page 8: Though the sustainability of MRT preconcentration with time has not yet been explicitly determined, we hypothesize that parameters such as the shape and volume of the sample reservoir and microcapillary could influence both the MRT duration limit and concentration rate.

2. Larger diameter capillary (Line 312): what is the influence of capillary dimensions on this phenomenon?

The statement made here was directly addressing the limitation of volumetric flow rate (Q) that was presented in the previous statement.

$$Q = v \cdot \pi \cdot \left(\frac{d}{2}\right)^2$$

At the same average flow velocity (v), a larger diameter (d) capillary would deliver more material proportional to the diameter squared, potentially enabling faster sample collection in the reverse-

mode MRT operation. However, the reviewer has raised a good question concerning the effect of capillary diameter on MRT in general, which could certainly have additional effects on the collection rate in the reverse-mode. We have done a few proof-of-concept experiments in larger diameter capillaries (see Supplementary Fig. 4 and Movies 3-6), but we have not yet done a thoroughly characterized the effects of capillary diameter and flow rate experimentally or using the numerical simulation. Thus, we have modified the statement to better indicate that we are addressing the volumetric flow rate limitation alone and acknowledge our lack of knowledge on the other effects this could have:

Page 8: Using a larger diameter capillary could help to overcome this volumetric flow rate limitation, although the relationship between capillary diameter, flow rate, and MRT concentration has not yet been tested.

3. What is/could be the influence of the DNA state (double-stranded versus single-stranded)? Could this technique be used for other molecules (proteins, charged polymers, etc.)?

We agree with the reviewer that these are significant questions, but we feel that they deserve a thorough investigation and their own manuscript. We can speculate that given the higher diffusivity of ssDNA that it could concentrate similarly or even more quickly than dsDNA, whereas proteins would be highly dependent on their charge. Moreover, different buffer systems could prove optimal for concentrating other molecular species (e.g. positively charged proteins and polymers). We hesitate to comment on this in this manuscript because it is purely speculation; however, we are also very interested in this topic and plan to pursue it in future research.

Minor comments:

1. There are a lot of undefined abbreviations, which may be obvious for some but not all readers.

Please define when first used. Some are listed below.

Line 48: define CE (capillary electrophoresis?)

Line 52: define EOF (electro-osmotic flow?)

Line 62: define HPLC, HDC (high-pressure liquid chromatography? Hydrodynamic chromatography?)

Line 74: what are "drag tags"?

Line 193: define EB

Line 195: define EACA, Bis-Tris

We apologize for the use of unfamiliar abbreviations and terminology without appropriate definitions. We have added definitions where appropriate and removed redundancies. We have also provided a more detailed description of the buffer reagents in the Reagent and Buffer Preparation subsection.

2. Line 217: where does 2000-fold come from? Fig. 5 shows 200.

We regret this oversight and thank the reviewer for bringing this mistake to our attention. The text has been amended to accurately refer to the > 100 fold concentration demonstrated in the figure:

Page 6: For the high electrophoretic force HCl buffer, concentration factors greater than 100 fold were achieved (Fig. 5, blue).

3. Line 443, 446: "The data/code that support the findings of this study are available from the authors upon reasonable request". What is a "reasonable request"? In my experience, either the data is shared (that is deposited and described on a public online site such as Figshare) or not. In the former case, the benefit is that the data is granted a DOI, which can then be used for traceability. In the latter case, opening the door to conditional individual requests will rapidly become bothersome to answer individual requests (not mentioning data loss, first author moving to a new position and unable to help, etc). The same goes for code, which can be easily deposited on Github, Bitbucket or similar sites (rather than institutional sites which usually do a poor job of preserving permanent links). Note that I am not asking for either data or code, because I have no time to spend digging into either now, but I would hate to be told that some of my requests are unreasonable if I did.

We appreciate the reviewer's candid comments on this important issue. We prefer that interested parties request the data/code from us directly so that we can assess the amount of interest for these materials. Thus, we used the suggested wording from Nature Research's Data Availability policy document (<https://www.nature.com/authors/policies/data/data-availability-statements-data-citations.pdf>). We certainly did not intend to be ambiguous or misleading. If it becomes too laborious to fulfill repeated requests, we will certainly take the reviewer's excellent suggestion to deposit the material on suggested websites.

4. There are a few typos, e.g. Line 291: "an limit of detection"

We thank the reviewer for pointing this out. We have rewritten this phrase (page 8) to read "a limit of detection" and have also perused the manuscript for any additional typos.

Reviewer #3 (Remarks to the Author):

The manuscript "Molecular Rheotaxis: Inducing DNA Migration and Concentration Against a Pressure-Driven Flow" by Friedrich et al, reports on a microfluidic preconcentration procedure for DNA, that is based on the well-known phenomenon, rheotaxis.

As mentioned, rheotaxis is a known concept (relating to organisms and particles, etc.), so the statement in the abstract that says 'This unexpected and counterintuitive migratory..,' and in the discussion ('..new, unexpected method..' are redundant. Why would it be surprising? Extraction of analytes under counter-current flow is also known.

If the authors wish this to attract more attention for those outside the community, perhaps they should decide on a more accessible term, other than molecular rheotaxis, for the procedure. Those interested in preconcentrating, say aqueous contaminants, in microfluidic systems, may find potential use of the approach.

We appreciate the reviewer's concerns regarding the term "Molecular Rheotaxis". We ultimately chose to call the phenomenon Molecular Rheotaxis because we felt it effectively and succinctly described the observed molecules' behavior in response to the flow stimulus (i.e. migration against a flow). As the reviewer pointed out, rheotaxis has been observed and described in not only fish, but other organisms (such as bacteria and sperm), as well as engineered particles. We have expanded our introduction to rheotaxis beyond fish to include bacteria, sperm cells, and engineered particles along with appropriate references:

Page 2: We have termed the phenomenon Molecular Rheotaxis (MRT), inspired by the biological rheotaxis, alignment and migration against a flow current, of fish²¹, bacteria^{22, 23}, and sperm cells²⁴, and the "artificial" rheotaxis of engineered anisotropic particles^{25, 26}.

The reason we found the result surprising, is that we are unaware of any other description of molecules undergoing migration against a flow (rheotaxis) in the absence of another externally applied force (such as an electric field). Without an understanding of the additional mechanisms described in the manuscript, one would not expect the migration and concentration behaviors observed in MRT. Due to the reviewer's concerns, we have revised the abstract (pasted farther below) and removed the phrase "unexpected and counterintuitive".

Generally, the work is interesting, it being about sample preconcentration in a microfluidic device, albeit only a single type of analyte, and could find applicability by those in these fields. One of the bugbears of microfluidic systems is that sample preparation (including cleanup, and enrichment (preconcentration)) is often still performed outside of the system, and needs additional efforts (involving in-house setups like those indicated by the authors in the introduction). Here, a potential inline approach (something that is integrated within the microfluidic system) is reported, that is seemingly easy to implement, may pave the way for truly practical use of micro-total analytical systems for real-world applications, although the broader appeal is unknown since the present work deals with DNA only (and it is not clear if the procedure works with mixtures of multiple components). It is also unclear if smaller molecules will be subjected to the same phenomenon.

We agree with the reviewer that DNA is only a small subset of aqueous analytes of interest, and analyzing the effects on a broader range of analytes, including small molecules and mixtures of molecules, will help to gauge the full utility and versatility of the technique. This is one topic we plan to pursue in future research.

Field-amplified sample stacking and isotachophoretic approaches are compared with the present one in terms of the limit of detection for DNA; one would have expected a more comprehensive evaluation these and other existing procedures.

We thank the reviewer for this perspective. We agree with the reviewer that an important aspect of describing this new concentration technique is to compare it with other existing methods. We had avoided an in-depth comparison with the range of preconcentration techniques because 1) we wanted the paper's focus to be a description of the science driving the underlying mechanism and 2) we did not want the paper to read as a review of in-line concentration techniques. Because the existing preconcentration techniques are molecular species-dependent (charge, molecular weight, etc), we felt it was more appropriate and meaningful to only compare our results with DNA preconcentration techniques since that was the subject of our study. Therefore, we chose to compare MRT-SML-FSHS with the most commonly used in-line DNA preconcentration and separation techniques: FASS-CE and ITP-CE. However, we have included citations to recent review articles that contain excellent comprehensive comparisons for the interested readers to access (13, 59, 60). We have also adjusted our wording to make it clear that we are comparing MRT-SML-FSHS to not just FASS-CE and ITP-CE strictly, but also to related techniques:

Page 8: DNA analysis with FASS-CE, ITP-CE, and related electrokinetic platforms can boast limits of detection in the range of low pM to tens of fM^{13, 59, 60}.

Generally, this work is publishable.

Minor comments:

The abstract should provide more quantitative data and information.

This is an excellent suggestion that can provide context to the readers. It has been rewritten as shown below:

This paper describes a microfluidic DNA preconcentration technique that does not require an externally applied electric field. Instead, pressure-driven flow from a fluid-filled microcapillary into a lower ionic strength DNA sample reservoir induces spontaneous DNA migration against the direction of flow and towards the capillary orifice. This migratory phenomenon that we call Molecular Rheotaxis (MRT) initiates quickly (< 5 seconds), can persist for long time periods (> 40 minutes), and results in a concentrated DNA bolus at the capillary orifice. We demonstrate that this preconcentration method acts on a wide size range of DNA (at least 2-27 kbp) and can be easily integrated with DNA size separation and single molecule detection as a highly sensitive microfluidic total analysis system. Paired experimental and numerical simulation results are used to delineate the parameters required to induce MRT, elucidate the underlying mechanism, and optimize conditions to achieve DNA concentration factors exceeding 10,000 fold.

Is it necessary to present results and data at the end of the introduction?

We are aware that the formatting of the introduction and results section is somewhat unconventional, but we felt that it was important to reveal the context in which this method was developed (i.e. by accidental observation). We felt that including these initial observations in the introduction, before revealing the mechanistic underpinnings, would help to convey the feelings of surprise and curiosity that captivated us when we first started to investigate this topic.

Why is the capillary rinsed for 'at least 11 min'? How is this value arrived at?

Eleven minutes corresponds to approximately 1.5 column volumes. We have added this description to the text within the methods section:

Page 11: The capillary was always rinsed with running buffer at least once between each separation for at least 11 minutes (approximately 1.5 column volumes).

Error bars are missing from the relevant plots.

We discuss the repeatability of SML-FSHS quantification and MRT-SML-FSHS in Supplementary Figure 11.

Additional comments:

In several instances, abbreviations are repeatedly defined.

We thank the reviewer for this observation. We have revised the manuscript to avoid repeated definitions of MRT and SML-FSHS without sacrificing clarity.

Check ref. 52: Author's name is completely capitalized; ref. 24, 22: Check the reporting of the reference; refs. 5, 10, 26, 29, 35, 36, 39, 40, 43, 55, 56: Check reporting format of respective journal titles. 28

We thank the reviewer for their attention to detail. We have made a concerted effort to correctly format every reference, including those listed above.

REVIEWERS' COMMENTS:

Reviewer #2 (Remarks to the Author):

The authors have satisfactorily addressed this reviewer's comments.

Reviewer #3 (Remarks to the Author):

This is a revised manuscript by Friedich et al that is being re-reviewed.

I believe the authors have covered all the questions and comments raised by all the reviewers, including myself, satisfactorily. Let me commend them for detailing their responses with such completeness and measuredness, and for indicating clearly where revisions in the manuscript have been made. As an editor myself, I often have to exhort authors to do exactly what the present ones have done.

I think the revised manuscript is suitable for Nature Communications now.